# Small-Molecule Fluorescent Probe for Detection of Sulfite

**DOI:** 10.3390/ph15111326

**Published:** 2022-10-26

**Authors:** Ting Li, Xuyang Chen, Kai Wang, Zhigang Hu

**Affiliations:** Medical Laboratory of Wuxi Children’s Hospital, The Affiliated Wuxi People’s Hospital of Nanjing Medical University, Qingyang Road 299, Wuxi 214023, China

**Keywords:** fluorescent probes, sulfite detection, small-molecule, reaction mechanisms

## Abstract

Sulfite is widely used as an antioxidant additive and preservative in food and beverages. Abnormal levels of sulfite in the body is related to a variety of diseases. There are strict rules for sulfite intake. Therefore, to monitor the sulfite level in physiological and pathological events, there is in urgent need to develop a rapid, accurate, sensitive, and non-invasive approach, which can also be of great significance for the improvement of the corresponding clinical diagnosis. With the development of fluorescent probes, many advantages of fluorescent probes for sulfite detection, such as real time imaging, simple operation, economy, fast response, non-invasive, and so on, have been gradually highlighted. In this review, we enumerated almost all the sulfite fluorescent probes over nearly a decade and summarized their respective characteristics, in order to provide a unified platform for their standardized evaluation. Meanwhile, we tried to systematically review the research progress of sulfite small-molecule fluorescent probes. Logically, we focused on the structures, reaction mechanisms, and applications of sulfite fluorescent probes. We hope that this review will be helpful for the investigators who are interested in sulfite-associated biological procedures.

## 1. Introduction

SO_2_ is one of the main pollutants in the environment, and SO_2_ derivatives are a collective term for SO_2_ in its state of existence in living organisms, mainly in the form of sulfite (SO_3_^2−^) and bisulfite (HSO_3_^−^) in a neutral environment [1,2]. Sulfites are widely used as an antioxidant additive and preservative in food and beverages [3,4,5,6]. On the other hand, in living organisms, sulfur-containing amino acids are metabolized to produce endogenous sulfites [7]. Abnormal endogenous or exogenous sulfite levels have been reported to be associated with certain diseases, such as asthma, dyspnea, chest tightness, respiratory arrest, anaphylaxis, diarrhea, hypotension, migraine, stroke, brain cancer, lung cancer, and liver cancer [8,9,10]. According to the Joint FAO/WHO Expert Committee on Food, the daily intake of sulfites is limited to 0.7 mg/kg body weight [11,12]. The U.S. Food and Drug Administration (FDA) has stipulated that the level of sulfites in food and beverages should not exceed 125 μM [13]. The Chinese Hygienic Standard for the Use of Food Additives stipulates that in terms of SO_2_ content, cookies, sugar, vermicelli, and canned foods shall not exceed 0.05 g/kg and other varieties shall not exceed 0.1 g/kg [14].

In recent years, numerous scholars have become interested in the effects of sulfites on living organisms. A study by Dalaman [15] et al. found that SO_2_ significantly prevented the prolongation of QT interval and action potential duration in rats after isoproterenol induction. SO_2_ derivatives inhibit the proliferation of human skin stratum corneum-forming cells by inhibiting the ERKI/2 and P38 pathways through activation of the NF–κB pathway [16]. Macrophage-derived SO_2_ is an important regulator of macrophage activation, and it acts as an endogenous “switch” in the control of macrophage activation [17,18]. Therefore, it is necessary to develop easy-to-operate, efficient, and inexpensive methods for the detection of sulfites. There are many methods for the detection of sulfites, such as chromatography, electrochemistry, capillary electrophoresis, flow injection analysis, and chemiluminescence [19]. However, these methods usually require complex procedures, and expensive instrumentation, and do not accurately assess sulfite fluctuations in biological systems. In contrast, fluorescence spectroscopy has the advantages of fast response, high sensitivity, simple operation, low cost, and in situ bioimaging within living cells for the detection of sulfites.

In the last decade, fluorescent probes have been used for the detection of various anions [20], such as ClO^−^, Cl^−^, NO_2_^−^, F^−^, CN^−^, S^2−^, Br^−^, etc [21,22,23,24,25,26,27]. In 2010, Choi et al. reported the first fluorescent probe for sulfite [28]. This research has attracted the attention of more and more researchers, and the construction of sulfite fluorescent probes is gradually becoming a popular research direction. Up to the present position, the number of research papers per year has increased dramatically, but there are few comprehensive reviews on small-molecule fluorescent probes for sulfite monitoring and applications. In this paper, we summarize the research progress of small-molecule fluorescent probes on sulfite in the last decade. In addition, this paper classifies sulfite fluorescent probes, including reaction mechanisms and constituent materials. We hope that this review will be useful for researchers who are interested in sulfite-related biological processes.

The design principle of sulfite fluorescent probes is related to their nucleophilic nature, where sulfite can undergo nucleophilic addition reactions with aldehyde groups or the deprotection of levulinic acid changes the probe structure causing changes in fluorescence spectra. A typical sulfite fluorescent probe consists of the three following main components: a fluorescent group, a recognition group, and a linker. Several important fluorophore of sulfite fluorescent probes are listed in this paper, including benzothiazole, coumarin, hemicyanine, quinoline, naphthalimide, benzimidazole, imidazole, triphenylamine, thiophene, pyrene, julolidine, Ir(Ⅲ) complex, naphthalimide, rhodamine, flavor, which are further subdivided according to the mechanisms.

At present, the response types of these fluorescent probes are mainly classified into “turn-on” and “ratiometric” strategies. With sulfite, the main mode of the turn-on strategies is the switching on and enhancement of the fluorescence signal. In recent years, more and more fluorescent probes are of the ratiometric strategies, because compared to single-emission turn-on probes, ratiometric probes can achieve precise detection by the ratio of the two emitted signals, effectively mitigating conditions, concentrations, and instrumentation using self-calibration. The ratiometric probes can achieve accurate detection by the ratio of the two emitted signals and effectively mitigate the interference of conditions, concentrations, and instruments by self-calibration [29,30].

## 2. Small-Molecule Sulfite Fluorescent Probe

### 2.1. Based on Benzothiazole Fluorophore

Benzothiazole prolongs cellular conjugation *π*, increases cellular water solubility, and contributes to the selective aggregation of probes in cellular mitochondria. Benzothiazole functionalization greatly improves the analytical performance by increasing the quantum yield, emission band red shift, and introducing a second emission band [31]. We classified benzothiazole fluorophore into the following categories based on the mechanism including intramolecular charge transfer (ICT), excited state intramolecular proton transfer (ESIPT), and fluorescence resonance energy transfer (FRET).

#### 2.1.1. ICT Mechanisms

ICT is the most common mechanism in fluorescent probes. The ICT mechanism refers to the process by which, in the excited state, the molecule will produce an electron transfer, resulting in the separation of the positive and negative charges. Generally, probes containing the ICT mechanism contain a push–pull system, i.e., “D−*π*–A” structure: where D (Donor) represents the electron donor and A (Acceptor) represents the electron acceptor, the electron transfer channel is provided by the *π* bond, and the final conjugated system is formed, and the electron donor or electron acceptor is the recognition group or part of the recognition group. When the probe reacts with the specific substrate, the electron donor or electron acceptor of the recognition group will change, resulting in a push-pull system and the *π* electron structure in the system is rearranged, resulting in the red shift or blue shift phenomenon of the fluorescence spectrum. We have developed probes **1**–**5** based on ICT mechanisms, of which probe **1** was the turn-on type and probes **2**–**5** were the ratiometric type. Probe **2** was reversible, probe **3** was mitochondrial-targetable, and probe **5** detects both F^−^ and SO_3_^2−^ (Figure 1).

Probes **1**–**3** are capable of visualizing the detection of HSO_3_^−^ and SO_3_^2−^. The addition of HSO_3_^−^/SO_3_^2−^ activates the Michael receptor of probe **1** [32], disrupting the *π*-electron conjugation system and changing the solution color from purplish red to colorless. Meanwhile, probe **1** was successfully applied to detect SO_3_^2−^/HSO_3_^−^ in real samples and HeLa cells. In contrast to probe **1**, probe **2** can reversibly monitor intracellular SO_3_^2−^/HSO_3_^−^ [33]. Probe **2** itself showed near-infrared fluorescence emission at 630 nm, and the addition of SO_3_^2−^/HSO_3_^−^ caused a gradual decrease in fluorescence intensity at 630 nm, but a significant enhancement of fluorescence at 630 nm was observed after the addition of H_2_O_2_, which was effective in restoring the system. To promote the application, the authors successfully used to monitor the redox process in cells and zebrafish. Compared with probe **2**, probe **3** [34] has a lower detection limit and enables the detection of SO_2_ derivatives in various environments. Moreover, a comparative analysis of snow water and industrial wastewater revealed severe SO_2_ contamination, and probe **3** has monitored the uptake, transport, and intracellular processes of exogenous HSO_3_^−^ within the HeLa cell successfully.

The large Stokes shift can well separate the excitation and emission light and improve the sensitivity of fluorescence detection while reducing interference. Probe **4** [35] showed a large Stokes shift before and after the addition of SO_3_^2−^, based on the large Stokes shift after excitation of the fluorophore of benzothiazole derivatives due to the ESIPT process. Previous studies—because the introduction of the sensing group into the hydroxyl group—inhibited the ESIPT, so that the Stokes shift was reduced, so probe **4** added an aldehyde group as the reactive site for SO_3_^2−^, so the measurement process before and after the addition of SO_3_^2−^ had a large Stokes displacement. Importantly, the probe has good selectivity for SO_3_^2−^ and biothiols.

Qi [36] et al. developed a ratiometric fluorescent probe **5** capable of detecting both F^−^ and SO_3_^2−^. Probe **5** has the two following sensing groups: the tert-butyldimethylsilyl ether moiety of F^−^ and the carbon–carbon double bond of SO_3_^2−^. Emission titration experiments performed found that the addition of F^−^ enhanced the fluorescence intensity ratio of the probe by 291-fold, while the addition of SO_3_^2−^ enhanced the fluorescence intensity ratio of the probe by 9445-fold. The authors suggest that the fluorescence is blue shifted by the weakening of ICT through Michael addition reaction between SO_3_^2−^ and carbon–carbon double bond. However, the proposed mechanism lacks a theoretical basis. Thus, Jia [37] and her team in a study published in 2021 revealed the detection mechanism of probe **5** for F^−^ and SO_3_^2−^ by density general function theory and time-varying density general function theory calculations, and the fluorescence red shift of the probe after the addition of F^−^ was caused by ESIPT, while the fluorescence blue shift of the probe after the addition of SO_3_^2−^ was caused by ICT.

#### 2.1.2. ESIPT Mechanisms

Excited state intramolecular proton transfer is the process of transferring intramolecular protons between a donor and an acceptor after the probe is excited. Molecules containing the ESIPT mechanism in general usually have amino and hydroxyl groups as proton donors, which are linked to neighboring proton acceptors (mostly S, N, and O atoms) through hydrogen bonds (Figure 2). After the proton transfer, a more stable molecule is formed and the molecular structure is usually changed from enol to ketone, and the fluorescence spectrum is changed accordingly. Generally, for substances that can undergo intramolecular proton transfer, there is often a significant Stokes shift in the fluorescence emission peak of the substance, which effectively reduces background interference.

Benzothiazole and its derivatives are the best-known ESIPT dyes, and the relative intensity variations of their photovariant isomers are often cleverly applied in the construction of scaled fluorescent probes. The emission of benzothiazole is usually confined to the blue and green regions and rarely exceeds 600 nm, and a straightforward way to obtain red shifted emission is to extend the *p*-conjugation of the ESIPT fluorophore. Zhang [38] et al. constructed probe **6** by combining benzothiazole with indole via C=C. This design extended the *p*-conjugated structure of benzothiazole and showed a larger red shift in both absorption and emission spectra compared with benzothiazole alone. After the addition of HSO_3_^−^, a rapid and significant change in the fluorescence spectrum was observed. Importantly, probe **6** can detect HSO_3_^−^ in food and HeLa cells. Probe **7** [39] has a similar structure and sensing mechanism to probe **6**, a conjugate of benzothiazole and semicarbazide. Probe **7** is more suitable for sensing intracellular SO_3_^2−^ and successfully detected SO_3_^2−^ in MCF-7 cells.

In 2021, Zhu’s [40] group constructed a benzothiazole-based probe **8**. The experimental results show that probe **8** has high application in HeLa cells and mice because it can detect endogenous SO_2_. Moreover, it provides a reliable non-invasive method for visualizing SO_2_ in mouse models during oxidative stress. More noteworthy is that probe **8** responds instantaneously (10 s) to sulfite.

#### 2.1.3. FRET Mechanisms

Fluorescent probes based on fluorescence resonance energy transfer generally contain two fluorescent groups (energy donor D and energy acceptor A, respectively). If D is used to excite the whole system, the transfer process of energy from D to A will occur, and finally, the fluorescence of A will be presented. The following several elements must be present in a probe with a FRET mechanism: (1) The fluorophore as the energy donor, whose fluorescence emission is generally at a short wavelength, and the absorption spectrum of the fluorophore as the acceptor must overlap the emission spectrum of the donor fluorophore so that the acceptor fluorophore can absorb the energy of the energy donor at the emission wavelength. (2) The collision diameter between the energy donor and the energy acceptor must be much smaller than the distance between the donor and the acceptor. (3) The arrangement of the energy donor and the energy acceptor must be appropriate (Figure 3).

A ratiometric fluorescent probe **9**–**11** was developed based on benzothiazole fluorophore and FRET mechanisms and was able to detect the amount of HSO_3_^−^ in MCF-7 cells. In 2014, probe **9** [41] was designed using benzothiazole and coumarin as fluorophores and was able to detect HSO_3_^−^ concentrations between 0.934 and 100.0 mM. The experimental results revealed that the ratio of fluorescence intensity when the probe reacted with HSO_3_^−^ increased continuously with increasing pH in the range of pH 5–8, which is due to the hydrolysis of coumarin under alkaline conditions. To overcome this obstacle, in 2016, Zhang [31] et al. designed probe **10** based on benzothiazole and anthocyanine dyes as fluorophores, which can react with HSO_3_^−^ in PBS solvent, but the response of probe **10** to HSO_3_^−^ takes 15 min, LOD = 340 nM. Wang’s [42] group optimized and introduced TCF with strong electron-absorbing properties to combine with benzothiazole, which can detect HSO_3_^−^ in the range of 5–40 μM, and the response time of probe **11** with HSO_3_^−^ is only 3 min with LOD = 101 nM (Figure 4).

### 2.2. Based on Coumarin Fluorophore

Coumarin was selected as a fluorophore due to its large Stokes shift, good solubility, high quantum yield, and excellent push–pull electron system. In the literature of the last 10 years summarized by me, the study of coumarin as a probe for fluorophores was mainly focused on 5 years ago. We classified coumarin fluorophore into the following categories based on the mechanism including intramolecular charge transfer (ICT), photon-induced electron transfer (PET), and fluorescence resonance energy transfer (FRET).

#### 2.2.1. ICT Mechanisms

Based on ICT mechanisms, probes **12**–**22** were developed. Among them, probes **13**, **16**, and **19** are turn-on probes, and probes **12**, **14**–**15**, **17**–**18**, and **20**–**22** are ratiometric. Probes **20**–**22** can localize to mitochondria, and probes **15**–**16** are mainly dedicated to distinguishing sulfites from sulfides.

In 2013, Sun [43] et al. designed probe **12** based on the addition-rearrangement cascade reaction, which reacted with sulfite and increased the emission ratio by about 1110-fold, but the probe could not react with HSO_3_^−^ in one step, the reaction time required 5 min, and the detection limit was 380 nM. Due to the slow response time, the maximum emission wavelength of the probe gradually changes, which is not suitable for real time detection of sulfite content in cells. So in 2015, Wang’s [44] team made an improvement and designed probe **13**, which incorporates an electron-withdrawing group (-CN) to accelerate the reaction with SO_2_ derivatives by activating carbon–carbon double bonds, so that the reaction is completed within 1 min.

The hydrophobic and alkaline microenvironment provided by the cationic surfactant micelles made the reaction of the probes with sulfite possible in an aqueous solution, thanks to the inspiration of Adamo and his colleagues’ research. Therefore, Tian and his team [45] designed and synthesized three kinds of probes **14** (as show in Figure 5) and used CTAB micelles to assist the reaction of the probes with sulfite.

Comparative experiments conducted found that the probe did not respond to sulfite after 20 h in the absence of CTAB and that the probe had poor selectivity (mainly HS^−^ and GSH had some effect on the detection) and a long reaction time. To solve the drawback of poor selectivity, next Tian and his team [46,47] designed probe **15** to distinguish sulfite from sulfide. Probe **15** covalently binds para-azidobenzenyl ketone to coumarin via a carbon–carbon double bond, distinguishing sulfite from sulfide by the different reaction sites (sulfite reacts with carbon–carbon double bond, sulfide reacts with azide group). Based on probe **14**, probe **16** adds m-pyridine and pyridine units to replace benzene, which enhances the electron-absorbing properties and can be distinguished by the spectral changes at different detection intervals for sulfites and sulfides; it also provides good water solubility, and compared with the previous two probes that require the assistance of CTAB micelles, probe **16** can complete the reaction under PBS solvent. To solve the drawback of long reaction time, Sun [48] et al. introduced an additional carbon–carbon double bond to construct probe **17** based on probe **14**, which extended the *π*-conjugation relationship, reacted with SO_3_^2−^, significantly promoted the nucleophilic incorporation rate, shortened the probe conjugation structure, and advanced the response time to 30 min (Figure 1).

Probe **18** [49] and probe **19** [50] are both reactive probes. Probe **18** uses Michael addition to dicyano-vinyl group, which has high selectivity for sulfite and active sulfur. The reaction is completed within the 30 s. The detection mechanism of probe **19** is a special nucleophilic addition reaction with aldehydes. The optimum pH of this probe for detecting HSO_3_^−^ is 5, which is not suitable for physiological environments. The detection limits of these two probes are too high, which limits their application.

Probes **20**–**22** are all mitochondrial-targetable. In 2015, Xu [51] et al. designed a probe with a detection limit of 90 nM and a response time of 3 min. This is the first probe that uses coumarin as a fluorophore and is localized to mitochondrial organelles. The probe not only showed a blue shift of 170 nm in the emission spectrum, but also two emission bands with good resolution after the addition of sulfite, which facilitated the fluorescence resolution (Figure 2). Yan’s team designed probes **21** and **22** in 2019 and 2020, respectively [52,53]. Both probes are near-infrared emission-mitochondria-targeted fluorescent probes based on FRET and ICT platforms. Probe **21** has a large Stokes shift, high photostability, chemical stability, and thermal cycling ability, and enables realizing the differentiation of endogenous L02 cells and cancer cells. Probe **22**, based on probe **21**, was used for exogenous detection of HeLa cells and endogenous detection of HepG2 and L02 cells, and the study yielded the result that the endogenous HSO_3_^−^/SO_3_^2−^ ratio detection can be successfully used to distinguish normal human cells and cancer cells. These mitochondria-targeting fluorescent probes have also made continuous breakthroughs in the detection limit.

#### 2.2.2. PET Mechanisms

Photoelectron transfer refers to a class of fluorescent probes in which the recognition group and the fluorescent group are the electron donor and the electron acceptor, respectively, and the process of electron movement from the donor to the acceptor occurs, and this process usually causes the loss of fluorescence. When the recognition group is not bound to the target molecule, the electrons of the fluorescent group will leap to higher energy orbitals after being excited by light, making the electrons of the recognition group rapidly replenish the orbitals of the fluorescent group, while the electrons of its fluorescent group disappear from fluorescence because they cannot return to their original orbitals. When the recognition group binds to the target molecule, the PET process cannot be completed due to the decrease in electron-donating capacity, and the electrons on the fluorescent group can return to their original orbit, and fluorescence is restored (Figure 6).

Probes **23**–**25** developed based on coumarin-benzopyridine fluorophore and PET mechanisms have the following advantages: good water solubility and mitochondrial-targeting, and all three probes are turn-on type fluorescent probes. Probes **23** and **24** have very similar structures, differing in that these two probes have different groups at the carbon 7 position. The hydroxyl-containing probe **23** [54] has a longer emission wavelength and shows strong fluorescence emission at 600 nm when reacting with HSO_3_^−^/SO_3_^2−^. The experimental results indicate that probe **24** is more suitable for the detection of HSO_3_^−^/SO_3_^2−^ in some water and food samples. Compared to probe **24**, probe **23** [55] can detect HSO_3_^−^/SO_3_^2−^ inside and outside the cell because probe **23** has a lower detection limit. It is worth mentioning that the probe responds instantaneously to fluorescence, and the fluorescence signal starts immediately after the addition of 0.4 equiv. of HSO_3_^−^ and can be stabilized within 15 s. On the contrary, the reaction time of probe **25** [56] with HSO_3_^−^ was 30 s and the detection limit was 42 nM, which was a mediocre performance compared to the other two probes.

#### 2.2.3. FRET Mechanisms

Ratiometric fluorescent probes **26**–**29** were designed based on FRET mechanisms. For biological applications, all of these probes can be used for intracellular imaging, and probe **28** can also be used for intra-mouse imaging. In terms of response time, probes **28** and **29** reacted with the SO_2_ derivatives for only a few periods. However, probes **26** and **27** required 1 h.

In 2015, Dai’s [57] group constructed the first two-photon ratiometric fluorescent probe **26** with a LOD of 53 nM in PBS buffer solution for single-photon excitation and 110 nm in PBS buffer solution for two-photon excitation.

In 2016, Zhao’s [58] group reported probe **27**, which has the potential to be used for differentiation of hepatocellular carcinoma cells and normal cells, incubated HepG2 cells and L02 cells with the probe, respectively, and found that only HepG2 showed significant changes in fluorescence after incubation with the probe. However, the response time of this probe to detect HSO_3_^−^ takes 1 h, limiting its application. To address this problem, Yang’s [59] group constructed probe **28**, the first donor structure to construct benzoic acid as a FRET fluorescent probe. The probe can reach stability with SO_3_^2−^/HSO_3_^−^ reaction within 3 min and has a Stokes shift of 239 nm. Importantly, probe **28** can detect SO_2_ derivatives in mice (Figure 3).

Recently, Ye’s [60] group investigated the metabolism of Cys to SO_3_^2−^/HSO_3_^−^ in mitochondria, and probe **29** is expected to be an effective tool for this purpose. Probe **29** detects SO_3_^2−^/HSO_3_^−^ by dual-channel imaging in MCF-7 cells. Besides, LOD = 26.3 nM, which is lower than all the probes mentioned above (Figure 7).

### 2.3. Based on Hemicyanine Fluorophore

Hemicyanine stems are a series of anionic probes containing anthocyanin dyes due to their long emission wavelength, potential over-fluorescence, specific binding sites, and good water solubility [61]. Hemicyanine fluorophores can capture nucleophilic reagents via 1,2—addition or 1,4—addition [62,63]. We classified hemicyanine fluorophore into the following categories based on the mechanism including intramolecular charge transfer (ICT), and fluorescence resonance energy transfer (FRET).

#### 2.3.1. ICT Mechanisms

Based on ICT mechanisms, probes **30**–**42** were exploited. In general, probes **31, 32, 38, 40, 42** were turn-on type, probes **30, 33**–**37, 39, 41** are ratiometric type.

In 2014, Sun [61] et al. developed probe **30** based on hemicyanine dye, which can detect HSO_3_^−^ in a 100% aqueous medium with a fast response time (within 90 s), and two well-separated emission peaks (Δλ = 106 nm) can be obtained before and after the addition of HSO_3_^−^, and they prepared a simple and rapid test paper for the detection of HSO_3_ ^−^.

In 2016, Samanta [64] et al. developed a multifunctional fluorescent probe **31** with highly sensitive dual recognition sensing of SO_3_^2−^ and SO_4_^2−^/HSO_4_^−^. The sensing mechanism of probe **31** for the anion SO_3_^2−^ is essentially due to the breakage of the conjugate extension limiting the ICT, which in turn leads to the fluorescence on response. In contrast, the presence of SO_4_^2−^/HSO_4_^−^ introduces the aggregation-induced emission (AIE) phenomenon to the system, which results in a turn-on fluorescence response. However, the detection limit of this probe was high (LOD = 106 nM). To further optimize it, in 2017, Yu [65] et al. developed a fluorescent probe capable of detecting SO_3_^2−^/HSO_3_^−^ and HSO_4_^−^ simultaneously using different emission channels. The detection limit of this probe was as low as 2.82 nM.

Because the positive charge of the anthocyanine derivatives can be localized to mitochondria, probes **33**–**36** had mitochondrial targeting. As shown by Figure 8, probe **33** [66] was a symmetrical semicarbazide structure that allows the probe to undergo two nucleophilic addition reactions with HSO_3_^−^, thus enabling the detection of large concentrations of HSO_3_^−^/SO_3_^2−^ and is the probe with the fastest response time (<30 s). In 2017, Yang [67] et al. developed two ratiometric probes **34a** and **34b** based on the electron-poor double bond structure. They studied the spectral properties of both probes and found that **34a** has an emission at 568 nm, while **34b** has an emission at 644 nm. This result leads to the conclusion that the emission spectra appear red shifted with the extension of the double bond. Additionally, the probe was used for fluorescence ratio imaging of endogenous HSO_3_^−^ in BT-474 cells to detect endogenous sodium bisulfite. Probe **35** [68] allows for real-time monitoring of SO_3_^2−^/HSO_3_^−^. In 2021, Lin [69] et al., based on the semicarbazide backbone, developed three probes **36a**, **36b**, and **36c** (see figure Figure 8). **36a** is a ratiometric probe with LOD = 0.27 mM and a response time of 50 min, and **36b** is a turn-on probe with LOD = 38.41 nM and a response time of 20 min, and the localization experiment found that the probe not only can localize to mitochondria but also have co-localization ability to Golgi apparatus, endoplasmic reticulum and other cell organelles.

In 2019, Qin’s group [70] developed a ratiometric fluorescent probe **37** that reacted with 500 μM SO_3_^2−^ and the color of the solution changed significantly from pink to colorless, probe **37** has the advantages of low LOD, high sensitivity, and high selectivity in HepG2 cells, and detects SO_3_^2−^/HSO_3_^−^ reaction very quickly (within 60 s). In contrast to probe **37**, probe **38** [71] applied HSO_3_^−^ fluorescence imaging in live mice, which was the first time used for imaging BALB/c mice to detect SO_3_^2−^/HSO_3_^−^ in live mice.

In 2020, the Wang’group [72] and Zhou’group [73] designed probes **39** and **40**, respectively, for bisulfite detection. Probe **39** is a ratiometric fluorescent probe that eliminates background interference and has a detection limit of 80 nM. The probe reacts rapidly with bisulfite and reaches stability within 2 min. In contrast to probe **39**, the turn-on probe **40** was able to monitor bisulfite in real time (reaction completed within 30 s), and, the probe was imaged in vivo in live mice, and the results suggest that the probe can be used for the detection of bisulfite in serum of live mice. Meanwhile, in the study of Pan et al. [74], probes **41a** and **41b** with the molecular structure as shown in the Figure 8 were designed. Both probes have a large Stokes shift (250 nm), and it was shown that **41a** containing carboxyl group responded more significantly to the fluorescence intensity change of HSO_3_^−^ in pure water than **41b** containing sulfoxide group, and **41a** has lower cytotoxicity and better biocompatibility compared to **41b** for the detection of HSO_3_^−^ in living cells, in addition to better response characteristics. However, when performing the optimization of the assay conditions, it was found that the assay was affected by the pH environment. To solve this problem, Shi [75] et al. designed to probe **42** with good acid and alkaline resistance within pH 3–11.

#### 2.3.2. FRET Mechanisms

Probes **43**–**46** were developed based on FRET mechanisms, where **43** is the turn-on type and **44**–**46** is the ratiometric type. The detection limit of probe **44** is 0.78 nM, the response time of probe **45** is 2 min, probe **46** has lipid droplet targeting, and probe **43** is a two-photon fluorescent probe.

In 2016, Zhu [76] et al. first reported a two-photon fluorescent probe **43** for imaging SO_2_ derivatives in biological tissues. The authors chose the acetyl fraction as the two-photon donor and the semicarbazone derivative as the quencher and recognition unit. Nuclear magnetic resonance spectroscopy and mass spectrometry demonstrated that the probe reacts with HSO_3_^−^ to disrupt the conjugate structure and the HOMO-LUMO energy gap of the semicarbazide derivative energy receptor increases, inhibiting the FRET process. Notably, probe **43** was successfully applied with the detection of HSO_3_^−^/SO_3_^2−^ imaging in HepG2 cells and rat liver tissue sections.

In 2018, Zhang’s [77] group constructed probe **44**, which detects exogenous SO_3_^2−^ in HeLa cells, and successfully detected cysteine metabolism in BRL cells. However, incubation of the probe with cells requires 4 h, limiting its application. Probe **45** [78] can detect HSO_3_^−^/SO_3_^2−^ “naked eye” with large Stokes shift (260 nm). However, the experimental results found that GSH, Hcy, and Cys can affect the detection.

Lipid droplets are dynamic subcellular structures of lipid metabolism, and lipid droplet abnormalities are associated with a variety of diseases such as obesity, fatty liver, and cardiovascular disease. Recently, Lin’s [79] group constructed probe **46**, which favors lipid droplets, the first probe constructed in concert with the FRET and ICT fluorescence platforms. Importantly, probe **46** successfully detected exogenous SO_2_ derivatives of HeLa cells and endogenous SO_2_ derivatives of HepG2 and L02, which are important for monitoring exogenous and endogenous HSO_3_^−^/SO_3_^2−^ and even diagnosing cancer cells.

### 2.4. Based on Quinoline Fluorophore

Tryptamine quinoline, as a typical ICT dye, has a large Stokes shift. Turn-on fluorescent probes **47**–**50** were designed based on quinoline and its derivatives, with probe **47** had mitochondrial-targeting and probe **49** had two-photon properties.

The mitochondria-targeted fluorescent probe **47** [80] has fluorescence emission in the near-infrared and is stable in the pH 3–10 range in response to SO_3_^2−^/HSO_3_^−^; however, the probe takes up to 2 h to enter the cell when detecting SO_3_^2−^/HSO_3_^−^ in HepG2 cells and zebrafish (Figure 4). It is known that background interference increases with longer detection cycles. In 2018, Xu’s group [81] designed probe **48** for this problem based on quinoline derivatives, which have an ester head and a dimethylamine tail in the structure of probe **48**, both of which have good cell permeability. Confocal fluorescence imaging performed to detect SO_3_^2−^ within HeLa cells showed that the probe was able to enter the cells within 2 min, largely shortening the detection time and reducing background interference. In addition, the authors also verified the low toxicity of probe **48** in HeLa, HEK293T, A549, and L02 cells of each cell.

In 2019, Zhang [82] et al. designed probe **49** to reveal the two-photon nature of tryptamine quinoline derivatives for the first time, and secondly, probe **49** introduces β-chlorovinyl aldehyde as a reaction site with the introduction of halogen, which retains the reaction properties of the aldehyde group with SO_2_ derivatives while introducing halogen for higher selectivity for thiol species. Compared with single photons, the two-photon penetration depth is stronger penetration, so the probe reacts with SO_2_ derivatives extremely fast (<5 s). To demonstrate the utility of the probe, the authors successfully used it for two-photon imaging of SO_2_ derivatives in live zebrafish.

In 2019, Zhou [83] et al. developed probe **50** using the mechanism of nucleophilic addition reaction of HSO_3_^−^ with *α,β*—unsaturated C=C. Probe **50** is a responsive fluorescent probe with a long emission wavelength of 598 nm and also has the advantages of fast response time, low detection limit, high sensitivity, and high selectivity (Figure 9).

### 2.5. Based on Naphthalimide Fluorophore

#### 2.5.1. ICT Mechanisms

Probes **51**–**54** were designed based on naphthalimide and its derivatives, where **52** is the turn-on type and **51** and **53**–**54** are the ratiometric types.

Hou’s group [84] and Zhang’s group [85] developed probes **51** using 4-hydroxynaphthalimide and levulinic acid in 2013 and 2014, respectively, and Hou investigated the sensing properties, selectivity, sensitivity, pH, and utility of the probes for HSO_3_^−^ in HEPES-buffered. Furthermore, Zhang et.al studied the spectral response, response time, sensitivity, selectivity, and utility of the probe for HSO_3_^−^ in ethanol and water (3:7). All proved that probe **51** is a good tool for the detection of HSO_3_^−^.

Abnormal SO_2_ levels further impair local immune function by affecting the amount and activity of lysosomal enzymes in macrophages, so it is important to monitor acidic lysosomal SO_2_ levels in biological systems. In 2017, Li’s group [86] proposed a fluorescent probe **52** with lysosomal targeting consisting of a morpholine unit, a semicyanine, and a naphthalimide fluorophore, and the probe **52** exhibits intense fluorescence emission at 524 nm only in the presence of both SO_2_ and H^+^. This is an AND-logic-based design concept. Besides, probe **52** has the advantages of high selectivity, fast response, and very low detection limit.

Reactive oxygen is a single-electron reduction product of molecular oxygen, and the dose, type, location, and duration of reactive oxygen species differ in the effects they cause on cells. HSO_3_^−^ is crucial as an antioxidant in regulating the balance of redox status in cells. Therefore, the monitoring of HSO_3_^−^/ROS (reactive oxygen) is of great interest. In 2021, probes **53** and **54** were proposed based on naphthalimide, respectively. Probe **53** designed by Wang [87] et al. is a reversible fluorescent probe that can be used to evaluate the redox state of HSO_3_^−^/H_2_O_2_ modulation in vitro and in vivo. Probe **53** itself can produce strong fluorescence emission at 580 nm, and after the C=C nucleophilic addition reaction with HSO_3_^−^, the fluorescence at 580 nm gradually decreases, and the strong fluorescence is emitted at 510 nm. Furthermore, the nucleophilic addition product can be oxidized by H_2_O_2_ to form the original C=C of the probe, and the fluorescence at 580 nm is re-enhanced. Moreover, HSO_3_^−^/H_2_O_2_ imaging was successfully performed in adult zebrafish and nude mice. Unlike probe **53**, probe **54** was proposed to identify HSO_3_^−^ and ClO^−^ by Wu [88] et al. Using different emission channels, probe **54** caused strong fluorescence emission at 515 nm when interacting with ClO^−^ and at 548 nm when interacting with HSO_3_^−^. In addition, the selectivity experiments showed that the probe has high selectivity and strong anti-interference ability. Notably, the authors successfully detected HSO_3_^−^ and ClO^−^ in plasma using probe **54** and performed intracellular imaging. The probe was also able to recognize endogenous ClO^−^ in vitro, thus distinguishing tumor cells and normal tissue cells.

#### 2.5.2. ESIPT Mechanisms

The C=N isomerization could be inhibited by an intramolecular N–H...N–C hydrogen bond, the formation of hydrogen bonds can limit intramolecular rotation and rigidify the molecular structure, thus helping to minimize the nonradiative energy loss of the exciton and maximize the probability of its radiative leap (open emission). This interaction has been successfully applied to the design of AIE and ESIPT fluorescent materials. Turn-on fluorescent probes **55**, **56** were designed based on ESIPT mechanisms.

In 2012, Sun’s [89] group reported for the first time the naphthalimide derivative probe **55** for the detection of HSO_3_^−^ content in white sugar, and the reaction products of probe **55** with HSO_3_^−^ were determined. Fluorescence titration and absorption titration experiments showed that probe **55** has good response properties. Recently, Huo [90] et al. constructed a probe **56** with the same structure, and probe **56** could detect SO_3_^2−^ in HeLa cells and mice while being able to localize to the lysosome with a response time of the 30 s (Figure 5).

### 2.6. Based on Benzimidazole Fluorophore

Probes **57**–**60** were designed based on benzimidazole and its derivatives, of which **60** were turn-on type and **57**–**59** were ratiometric type.

Back in 2011, probe **57** [91] was designed based on the selective reaction of SO_3_^2−^ with aldehydes. The reaction of probe **57** with SO_3_^2−^ can be stabilized within 5 min, and the detection range for SO_3_^2−^ is 2−200 μmol/L. Earlier developed probes had many shortcomings, and Wang et al. studied the sensing response of the probe to bisulfite in an ethanol and acetate buffer solution at pH 4.6 (20 mmol/L), thus limiting the detection of probe **57** in a physiological environment. Thus, after continuous progress, Niu [92] et al. designed probe **58** for DMF/PBS at pH = 7.4, and probe **58** emits at 664 nm, which is due to its molecular structure of 2-(2-hydroxy-phenyl)-benzoimidazole(HBN)-CHO and TCF (2-dicyanmethylene-3-cyano-4,5,5-trimethyl-2,5dihydrofuran) combined by unsaturated C=C bonds, which can red shift emission, and importantly, probe **58** successfully detected BEL-7402 intracellular and exogenous SO_2_ derivatives.

In 2019, Kavitha’s group [93] designed a probe **59** using phenanthrene imidazole as a fluorescent group with the advantages of high selectivity and fast response. The authors concluded by fluorescence, ^1^H NMR, and ESI-mass spectrometry studies that the mechanism of interaction of probe **59** with SO_3_^2−^ is based on the nucleophilic addition of carbon-carbon double bonds, blocking the *π*-conjugation. Moreover, the probe successfully monitored SO_3_^2−^ in real samples and HeLa cells.

The dicyanovinyl group is a group with strong electron-absorbing properties, which can reduce the electron density of the carbon–carbon double bond and therefore is easy to undergo nucleophilic addition reactions. Probe **60** [94] was developed based on the dicyanovinyl group and the maleimide moiety to separate bisulfite and thiol amino acids. The study examined the absorption spectra and fluorescence emission spectra of the probe with sodium bisulfite and thiol amino acids, and both observed changes in the spectra, but differed in the time at which the changes occurred; these changes were observed approximately 30 min after the addition of thiol amino acids, whereas in the case of sodium bisulfite, the changes were observed immediately. More importantly, probe **60** measures the level of HSO_3_^−^ in HeLa cells (Figure 10).

### 2.7. Based on Imidazole Fluorophore

The imidazopyridine derivatives have good photophysical properties, large Stokes shift, good photostability, and high fluorescence quantum yields. In 2019, Chen [95] et al. synthesized turn-on probes **61** using imidazopyridine derivatives as fluorescent agents and malononitrile moieties as SO_3_^2−^ reactive sites, probe **61** has the advantages of large Stokes shift, high sensitivity degree, and high selectivity, and successfully determined SO_3_^2−^ in MCF-7 cells and zebrafish (Figure 6).

### 2.8. Based on Triphenylamine Fluorophore

The ratiometric fluorescent probe **62** [96], a dual-channel chemosensor with both fluorescence and colorimetric response, is designed with an electron-rich triphenylamine-thiophene as the fluorophore and electron-donating group and an aldehyde group as the electron acceptor. With the increase of HSO_3_^−^ concentration, the maximum emission wavelength of probe **62** was shifted from 560 nm to 440 nm, and at the same time, the color changed from yellow to colorless. It is worth mentioning that the experimental results demonstrate that the addition reaction of an aldehyde with HSO_3_^−^ is reversible.

### 2.9. Based on Thiophene Fluorophore

Thiophenes are ideal building blocks for the synthesis of conjugated *π*-systems [97]. In 2021, Wang [98] et al. synthesized probes **63** with thienyl-substituted diketopyrrolopyrrole. They investigated HSO_3_^−^ detection and imaging by probe **63** in normal hepatocytes and hepatocellular carcinoma cells, and found that there is a difference in endogenous HSO_3_^−^ production by HepG2 and L02 cells, and the concentration of endogenous HSO_3_^−^ in HepG2 cells is much higher than that in L02 cells, which is important for the diagnosis of hepatocellular carcinoma.

### 2.10. Based on Pyrene Fluorophore

Pyrene and its derivatives are important compounds with a huge conjugation system and stable photophysical properties [99,100]. Therefore, they are often used as fluorophores to design fluorescent probes for the recognition of anions, cations, and neutral small molecules.

In 2021, Chao’s [101] group constructed a probe **64** based on pyrene derivatives. The results of mass spectrometry and infrared spectroscopy experiments showed that the reaction mechanism of probe **64** with SO_3_^2−^ was the nucleophilic attack of unsaturated carbon–carbon double bonds in the *α,β*−keto structure by SO_3_^2−^. More notably, endogenous SO_3_^2−^ could be detected in HepG2 cells.

### 2.11. Based on Julolidine Fluorophore

In 2018, Yao’s [102] group designed a ratiometric fluorescent probe **65** that successfully detected SO_3_^2−^ within HepG2 and L929 cells and was able to localize to mitochondria. The experimental results show that probe **65** detects SO_3_^2−^ in a two-step reaction, first adding in the C=N bond to shorten the conjugate structure of the probe, and then an intramolecular rearrangement occurs to further shorten the conjugate structure, resulting in a significant blue shift in the absorption and emission spectra. In 2019, Tamima’s [103] group designed a ratiometric fluorescent probe **66** that successfully detected SO_2_ derivatives in HeLa cells and was able to localize them to lysosomes. Probe **66** is based on the linear shape of the benzopyran dye system, which is a *π*-extended pyrroline system. Probe **66** has the following advantages: emission in the near-infrared wavelength range, the ability of the morpholine group in the structure to act as a partial quencher of PET, two-photon excitation, fully spectrally separated scale imaging, high sensitivity, and selectivity.

### 2.12. Based on Ir(III) Complex Fluorophore

Due to the high photoluminescence efficiency, long lifetime, and significant Stokes, bis-cyclometalated Ir(III) complex are increasingly used for the detection of analytes. In 2018, Gao’s group [104] constructed the turn-on fluorescent probe **67**. Probe **67** has the typical characteristics of Ir(III) complexes, and upon interaction with HSO_3_^−^, the probe absorbs at 377, 408, and 466 nm. Meanwhile, probe **67** can selectively detect HSO_3_^−^, and the experimental results found that HSO_3_^−^ is mainly present as SO_3_^2−^ when the pH is close to neutral, but SO_3_^2−^ does not respond to the probe.

### 2.13. Based on Rhodamine Fluorophore

In 2019, Liu’s [105] group reported a rhodamine probe **68** based on the Michael addition reaction. Probe **68** enables the detection of SO_3_^2−^ levels in HepG2 cells. The probe itself had two emission peaks at 450 nm and 566 nm, respectively, and the fluorescence intensity at 566 nm gradually decreased with the addition of SO_3_^2−^ and remained unchanged at 450 nm. The response time of this probe to sulfite is completed within the 30 s, which is better than some of the fluorescent probes reported so far.

### 2.14. Based on Flavor Fluorophore

In 2015, Xu [106] et al. constructed a probe **69** capable of detecting HSO_3_^−^ and Al^3+^ simultaneously, enabling the simultaneous detection of negative ions and cations by a single probe. However, the results of competitive and selective experiments show that Fe^3+^, Cu^2+^, and Mn^2+^ have some influence on the detection and poor selectivity. This design concept has great significance, but it needs continuous improvement (Figure 11).

## 3. Conclusions and Prospect

People are increasingly concerned about diet and health, and sulfite, as a common food antioxidant additive and preservative, can also be produced endogenously and is closely related to peoples’ daily life, so it is necessary to develop a simple, efficient, and inexpensive testing instrument. In the past decade, sulfite fluorescent probes have developed rapidly. This paper reviews the research progress of sulfite small-molecule fluorescent probes in the last decade.

We found that the probes constructed based on the ICT mechanism are the most basic and extensive. Researchers are continuously studying the composition of fluorescent groups with recognition groups and linkage bonds. For the first 5 years, mostly single-emission turn-on fluorescent probes were also used only to detect sulfite components in real samples (e.g., water, sugar, wine). In the latter 5 years, ratiometric fluorescent probes have been continuously developed for their advantages. More and more excellent fluorophores were also explored and gradually applied to image endogenous sulfites in cells, tissues, zebrafish, and mouse models. In addition, fluorescent probes with organelle (e.g., mitochondria, lysosomes, lipid droplets) targeting, two-photon, and reversible cycling have been introduced. Importantly, the response time of fluorescent probes for sulfite reactions is getting shorter, the detection limits are getting lower, and the water solubility is getting better. More notably, in the past two years, scholars were keen to develop red or near-infrared ratiometric fluorescent probes with excellent analytical properties such as low background interference, low biological damage, and deep tissue penetration, which are more suitable for in vivo imaging. However, a sulfite fluorescent probe that combines all the advantages is still being explored. Therefore, there are still the following challenges and difficulties in design and application: (1) When imaging sulfites in cells or tissues, the probe incubation time is slightly longer (>30 min), which is not conducive to application for real time monitoring in vivo. (2) The components in serum are complex and low, and the probes summarized do not currently detect sulfite concentrations in serum.

Combining the above advantages and disadvantages of sulfite small-molecule fluorescent probes, we believe that there is still much room for progress and development in this field. Future goals can be devoted to the construction of optical imaging of sulfite in vivo with low detection limits, high selectivity, no damage, stability in physiological environments, reversible cycling, and can be monitored in real time and used for sulfite.

## Data Availability

No new data were created or analyzed in this study. Data sharing is not applicable to this article.

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
