# Peer review of "Small-Molecule Fluorescent Probe for Detection of Sulfite"

_pharmaceuticals, 2022, doi:10.3390/ph15111326_

Round 1

Reviewer 1 Report

The review article submitted by Ting Li et al. entitled Small molecule fluorescent probe for detection of sulfite introduces the research progress of small molecule fluorescent probes for sulfite and bisulfite in the last decade. The authours focused on the structures, reaction mechanisms, and applications of sulfite fluorescent probes through 69 examples. Finally, the advantages and disadvantages of fluorescent sulfite probes and the future development direction are pointed out. This review is helpful for the investigators who are interested in sulfite-associated biological procedures. I think this review is a comprehensive survey and recommend the acceptance of this work with minor corrections.

The classification in the article is a little verbose, I suggest that the probe be classified according to the fluorophore containing. If the same fluorophore has several important mechanisms, then subdivide.

Reviewer 2 Report

This manuscript is an interesting summary of " Small molecule fluorescent probe for detection of sulfite".

This manuscript is acceptable in Pharmaceuticals after satisfactory revision of the following minor points.

1) Page 2 line 11, In the last decade, fluorescent probes have been used…. but there are few comprehensive reviews on small-molecule fluorescent probes for sulfite monitoring and applications, Please cite the few comprehensive reviews.

2) Page 2, If possible, it would be better to include a schematic diagram of the mechanism. ie. ICT, ESIPT, PET, FRET.

3) Page 2, …the following categories based on the fluorophores including benzothiazole (HBT),….

Benzothiazole is not HBT.  HBT is 2-(2-Hydroxyphenyl)benzothiazole.

4) Page 4, …. So in 2015, Wang's [37] team made an improvement and designed probe 7, which incorporates an electron-absorbing group (-CN) .  “electron-absorbing group” (-CN) should be “electron-withdrawing group”

5) Page 4, …. Probe 12 [42] and probe 13 [43] are both reactive probes. Probe 12 uses Michael to add dicyano vinyl, which has high selectivity for sulfite and active sulfur. “uses Michael to add dicyano vinyl” should be “Michael addition to dicyano-vinyl group”

6) Page 6, Scheme 1. captiom “The structures with typical detection mechanisms of sulfite fluorescent probes 1-16”.  The mechanism is not clear for compound 6.

7) Page 6, Hemicyandiamide fluorophores can capture nucleophilic rea- gents via 1,2-addition or 1,4-addition [48,49].  Can not found Hemicyandiamide fluorophores in ref. 48,49. “Hemicyandiamide” should be Hemicyan. Must be cite original paper.

7) Page 7, In 2017, Yang [53] et al. developed two ratiometric probes 21a and 21b based on the electro-poor double bond structure. “electro-poor double bond” should be “electron-poor double bond”.

8) Page 8, Scheme 2. Caption. The structures with typical detection mechanisms of sulfite fluorescent probes 17-29. No “detection mechanism” found in Scheme 2.

9) Page 8,  In Scheme 2, there are two positive charge in the structure of compound 19.

10) Page 9, Table 2019. Zhang [64] et al. designed probe 32 to reveal the two-photon nature of tryptamine quinoline derivatives for the first time, and secondly… What is “Table 2019” ? 

11) Page 9, Hou’group [66] and Zhang’group [67] developed probes 34 using 4-hydroxynaph- thalimide and levulinic acid in 2013 and 2014, “Hou’group” “Zhang’group” should be “Hou’s group” “Zhang’s group”.

12) Page 12, Scheme 3. Caption. The structures with typical detection mechanisms of sulfite fluorescent probes 30-42. No “detection mechanism” found in Scheme 3.

13) Page 13, Scheme 4. Caption. The structures with typical detection mechanisms of sulfite fluorescent probes 43-45.  Should be change the caption. All captions are same.

14) Page 17, Scheme 5. Caption. The structures with typical detection mechanisms of sulfite fluorescent probes 46-58Should be change the caption. All captions are same.
